# Doubly Robust Kernel Statistics for Testing Distributional Treatment Effects

**Jake Fawkes**  *jake.fawkes@st-hughs.ox.ac.uk*
*Department of Statistics*
*University of Oxford*

**Robert Hu**  *robyhu@amazon.co.uk*
*Amazon*[†]

**Robin J. Evans**  *evans@stats.ox.ac.uk*
*Department of Statistics*
*University of Oxford*

**Dino Sejdinovic**  *dino.sejdinovic@adelaide.edu.au*
*School of Mathematical Sciences*
*University of Adelaide*[†]

## Abstract

With the widespread application of causal inference, it is increasingly important to have tools which can test for the presence of causal effects in a diverse array of circumstances. In this vein we focus on the problem of testing for *distributional* causal effects, where the treatment affects not just the mean, but also higher order moments of the distribution, as well as multidimensional or structured outcomes. We build upon a previously introduced framework, Counterfactual Mean Embeddings, for representing causal distributions within Reproducing Kernel Hilbert Spaces (RKHS) by proposing new, improved, estimators for the distributional embeddings. These improved estimators are inspired by doubly robust estimators of the causal mean, using a similar form within the kernel space. We analyse these estimators, proving they retain the doubly robust property and have improved convergence rates compared to the original estimators. We then use the proposed estimators as test statistics in a new permutation based test for distributional causal effects. Finally, we experimentally and theoretically demonstrate the validity of these tests.

## 1 Introduction

In this work we focus on the problem of testing for distributional treatment effects (Bellot & van der Schaar, 2021; Park et al., 2021; Chikahara et al., 2022), where the aim is to test for causal effect which manifest as something other than a mean shift. This can be especially useful when the target variable is high dimensional or structured as a network, since in these cases there is no natural mean to compare. Our contributions are as follows:

1. We introduce new estimators to be used within the Counterfactual Mean Embeddings framework, based upon the doubly robust estimator of the causal mean from semi-parametric statistics (Bang & Robins, 2005; Tsiatis, 2006). These may be applied to estimate kernelised versions of the average treatment effect and effect of treatment on the treated. We prove these estimators inherit the double robustness properties of well established semi-parametric estimators of causal effects and that they converge to the correct value if either of the models underlying them converge to the true model. This shows that they have theoretically improved convergence results when compared to the previous counterfactual mean embedding estimators.

---

[†]Work mainly done while the authors were with the Department of Statistics, University of Oxford.

2. We apply these new estimators to permutation testing for distributional causal effects. We propose a new permutation approach which allows for doubly robust trainable statistics to be used within permutation testing and prove that these tests are valid.

3. We experimentally validate the performance of our test on synthetic, semi-synthetic and real data. An implementation of our approach can be found at: https://github.com/Jakefawkes/DR_distributional_test.

## 1.1 Related work

Our work builds upon the counterfactual mean embeddings framework introduced in Muandet et al. (2021), which we detail further within Section 2.3. This falls under the general area of applying kernels to test for distributional causal effects of which Bellot & van der Schaar (2021) is an early example, whose test statistic arises as a special case of our own for the distributional effect of treatment on the treated. In a concurrent work, Martinez-Taboada et al. (2023) develop a similar approach using doubly robust statistics (Robins & Rotnitzky, 1995) within kernel spaces. However, by applying the work of Shekhar et al. (2022) they are able to take a permutation free approach to testing distributional causal effects. Due to the concurrency and current lack of publicly available code we do not compare against this method. This work has also been built to estimate counterfactual densities in Martinez-Taboada & Kennedy (2023). In more general applications of kernel spaces to testing distributional causal effectsPark et al. (2021) develop a statistic targeting the conditional average treatment effect (CATE) using conditional mean embeddings to estimate the RKHS distance between the expected potential outcomes. Chikahara et al. (2022) use tests for distributional causal effects to find which features are relevant to difference in treatment effects. Testing for causal distributional effects falls within long tradition of using kernel spaces for hypothesis testing as they faithfully capture all features of a distribution (Gretton et al., 2005; 2012). Kernel methods have also been widely applied within the larger causality literature. For example, to instrumental variables (Singh et al., 2019; Muandet et al., 2020), to causal learning with proxies and unmeasured confounders (Singh et al., 2020; Mastouri et al., 2021), and to the orientation of causal edges (Mitrovic et al., 2018).

## 2 Background and Notation

### 2.1 Causal Set Up

Throughout, we will let $Y$ denote the outcome, which is affected by a binary treatment $T$ in the presence of additional covariates $X$. We will use the potential outcome notation (Rubin, 1997) so that $Y(t)$ corresponds to the outcome observed when $T$ is given the value $t$. We assume that the causal relationships between these variables are given by the basic confounding DAG in Figure 1 [1].

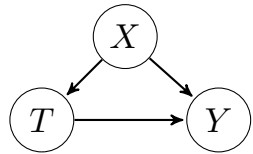

Figure 1: Assumed DAG

**The Propensity Score and Overlap**  Throughout we will make reference to the *propensity score*, which is the probability of treatment given a set of covariates: $P(T = 1 \mid X = x)$. For a flexible notation we will use $e(x, t)$ to denote $P(T = t \mid X = x)$, however as we are using binary treatment we will have that $e(x, t) = 1 - e(x, t')$ if $t + t' = 1$. We will also make use of the inverse propensity odds which we will denote $w(x, t) = \frac{1 - e(x, t)}{e(x, t)}$. Finally $\hat{e}(x, t)$ and $\hat{w}(x, t)$ will be used to denote finite sample estimates of these quantities.

An important assumption for causal inference from observational data is that of **overlap**. This ensures that there are treated and untreated individuals we can compare and it written formally as:

**Definition 1.** *We say there is **overlap** for $T = t$ if we have $0 < e(X, t)$ with probability 1. We say the overlap is **two-sided** if it holds for both $t$ and **one-sided** otherwise. Any overlap is **strict** if there is some $\delta > 0$ such that $\delta \leq e(X, t)$ again with probability 1.*

Under the one-sided overlap assumption that $e(x, t) > 0$ and the causal DAG in Figure 1, the distribution of the interventional quantity $Y(t)$ is uniquely identified (Pearl, 2009; Richardson & Robins, 2013) from

---

[1]We make us of the SWIG framework to combine causal graphical models with potential outcomes. More details can be found in Richardson and Robins Richardson & Robins (2013)

observational data as follows:

$$P(Y(t) = y) = \mathbb{E}_{P(X)}\left[P(Y = y \mid X, T = t)\right].$$

This allows us to estimate causal quantities of interest from observational distributions alone, such as the average treatment effect (ATE):

$$\mathbb{E}[Y(1) - Y(0)].$$

It is important to note that both-sided overlap is required for the identification of the average treatment effect, as each interventional distribution requires overlap at a different $T = t$ for identification. In the case where we are only likely to have one-sided overlap, such as when an experimental new medical treatment is only given to the most severe cases, practitioners instead focus on the effect of treatment on the treated (ETT):

$$\mathbb{E}[Y(1) - Y(0) \mid T = 1].$$

This quantity is identified from observational data whenever $e(x, 0) > 0$ for almost all $x$, so all individuals have some probability of not being treated. This allows us to still get some estimate of treatment efficacy in cases where both-sided overlap is violated. It is also worth noting that overlap is a testable assumption, with a test developed for example in Lei et al. (2021).

### 2.2 Kernel Background

We use kernel embeddings of distributions to construct our test statistics and so we now briefly introduce the relevant background material and notation. For a more thorough engagement with this material we refer the reader to Muandet et al. (2017).

**Reproducing Kernel Hilbert Spaces** Let $\mathcal{X}$ be some non-empty space. A real-valued RKHS, $(\mathcal{H}_{\mathcal{X}}, \langle \cdot, \cdot \rangle_{\mathcal{H}_{\mathcal{X}}})$, is a complete inner product space of functions $f : \mathcal{X} \to \mathbb{R}$ such that the evaluation function is continuous for all $x \in \mathcal{X}$. Due to the Riesz representer theorem we have that for all $x \in \mathcal{X}$ there is a function $k_x \in \mathcal{H}_{\mathcal{X}}$ which satisfies the *reproducing property*, that $f(x) = \langle f, k_x \rangle$ for all $f \in \mathcal{H}_k$. The function $k(x, x') = \langle k_x, k_{x'} \rangle$ is known as the reproducing kernel for the space $\mathcal{H}_{\mathcal{X}}$. Conversely, via the Moore-Aronszajn Theorem (Aronszajn, 1950), any symmetric positive definite function $K$ on $\mathcal{X}$ defines a unique RKHS. We will suppose access to an RKHS on $\mathcal{X}$ and $\mathcal{Y}$ denoted by $\mathcal{H}_{\mathcal{X}}$ and $\mathcal{H}_{\mathcal{Y}}$, with with kernels $k$ and $\ell$ respectively.

Further, for a random variable $X$ on $\mathcal{X}$ with distribution $P_X$ we can define the *mean embedding* of $X$ as:

$$\mu_X = \mathbb{E}_{P_X}[k(X, \cdot)].$$

Under the intergrability condition that $\int_{\mathcal{X}} \sqrt{k(x, x)} dP_X(x) < \infty$ we have that $\mu_X \in \mathcal{H}_{\mathcal{X}}$. If we have two random variables, $X, X'$, over $\mathcal{X}$ with distributions $P_X, P_{X'}$ respectively, with a slight abuse of notation we denote the *maximum mean discrepancy* (MMD) by:

$$\text{MMD}[X, X', \mathcal{H}_{\mathcal{X}}] := \left\| \mu_{P_X} - \mu_{P_{X'}} \right\|_{\mathcal{H}_{\mathcal{X}}}.$$

The kernel $k$ is known as *characteristic* if $\text{MMD}[X, X', \mathcal{H}_{\mathcal{X}}] = 0$ if and only if $P_X \overset{d}{=} P_{X'}$. Many popular kernels such as the Gaussian and Matérn are charecteristic kernels. The MMD between two distributions can be estimated from finite samples by computing the MMD between the empirical distributions. This was shown in Gretton et al. (2012) to converge to the true MMD at the paramteric convergence rate, $\mathcal{O}_P(n^{-\frac{1}{2}})$. For this reason the MMD is often used for efficiently testing equality of distributions.

**Conditional Mean Embedding** RKHSs also allow us to represent conditional distributions through the *Conditional Mean Embedding* (CME) (Song et al., 2009; Muandet et al., 2017), given by:

$$\mu_{Y|X=x} = \mathbb{E}[\ell(Y, \cdot) \mid X = x].$$

In order to estimate this Grünewälder et al. (2012) propose to take a regression point of view, seeing the CME as the solution to the following regression problem:

$$\begin{cases} C^* = \arg\min_{C \in B_2(\mathcal{H}_\mathcal{X}, \mathcal{H}_\mathcal{Y})} \mathbb{E}_{P_{(Y,X)}} \|\ell(Y, \cdot) - C k(X, \cdot)\|_{\mathcal{H}_\mathcal{Y}} \\ \mu_{Y|X=x} = C^* K(x, \cdot) \end{cases},$$

where $B_2(\mathcal{H}_\mathcal{X}, \mathcal{H}_\mathcal{Y})$ is the space of Hilbert-Schmidt operators $\mathcal{H}_\mathcal{X} \to \mathcal{H}_\mathcal{Y}$. This interpretation leads to an estimate of the CME from a finite dataset, $\mathcal{D} = \{x_i, y_i\}_{i=1}^n$ as:

$$\begin{cases} \hat{C}^* = \underset{C \in \mathsf{B}_2(\mathcal{H}_\mathcal{X}, \mathcal{H}_\mathcal{Y})}{\arg\min} \frac{1}{n} \sum_{i=1}^n \|\ell(y_i, \cdot) - C k(x_i, \cdot)\|_{\mathcal{H}_\mathcal{Y}}^2 + \gamma \|C\|_{\mathsf{B}_2}^2 \\ \qquad = \boldsymbol{\ell}^\top \mathbf{W} \mathbf{k} \\ \hat{\mu}_{Y|X=x} = \boldsymbol{\ell}^\top \mathbf{W} \mathbf{k}(x) \end{cases}. \tag{1}$$

where $\boldsymbol{\ell} = (\ell(y_i, \cdot))_{i=1}^n$, $\mathbf{k} = (k(x_i, \cdot))_{i=1}^n$, and $\mathbf{W} = (\mathbf{K} + \lambda \mathbf{I}_n)^{-1}$ for $\mathbf{K} = (k(x_i, x_j))_{i,j=1}^n$. We will often make use of the CME conditional on a particular value of $T$, so $\mu_{Y|X=x, T=t}$. In that case we let $\boldsymbol{\ell}_t$, $\mathbf{W}_t$ and $\mathbf{k}_t$ denote the respective matrices used in the estimation of $\hat{\mu}_{Y|X=x, T=t}$.

## 2.3 Counterfactual Mean Embeddings

Our work builds upon the counterfactual mean embeddings framework of Muandet et al. (2021). They demonstrate that under the causal structure in Section 2.1 and with the one-sided overlap $e(x, t) > 0$, the mean embedding of the potential outcome, $Y(t)$, can be written as:

$$\mu_{Y(t)} = \mathbb{E}_{P_{(Y,X,T)}} \left[ \frac{\ell(Y, \cdot) \mathbb{1}\{T = t\}}{e(X, t)} \right],$$

which may be seen as the kernel equivalent of the inverse probability weighting estimator for the causal mean. Given a finite sample, $\{(t_i, x_i, y_i)\}_{i=1}^n$, and access to the true propensity score, this quantity can be estimated as:

$$\hat{\mu}_{Y(t)} = \frac{1}{n} \sum_{i=1}^n \frac{\ell(y_i, \cdot) \mathbb{1}\{t_i = t\}}{e(x_i, t)}.$$

Muandet et al. (2021) prove that this converges to the true embedding at rate $\mathcal{O}_P(n^{-\frac{1}{2}})$. If we have both-sided overlap both $\hat{\mu}_{Y(1)}$ and $\hat{\mu}_{Y(0)}$ are therefore identified from data and we can use $\mathrm{MMD}[Y(1), Y(0), \mathcal{H}_\mathcal{Y}]$ to measure any distributional effects of treatment. Furthermore, if the kernel is characteristic we have that $\mathrm{MMD}[Y(1), Y(0), \mathcal{H}_\mathcal{Y}] = 0$ if and only if $Y(1) \overset{d}{=} Y(0)$. Finally, this quantity can be estimated from finite samples at rate $\mathcal{O}_P(n^{-\frac{1}{2}})$ by:

$$\widehat{\mathrm{MMD}}[Y(1), Y(0), \mathcal{H}_\mathcal{Y}] = \left\| \hat{\mu}_{Y(1)} - \hat{\mu}_{Y(0)} \right\|_{\mathcal{H}_\mathcal{Y}}.$$

This leads to a permutation test for distributional causal effects following the conditional permutation scheme for testing for causal effects (Rosenbaum, 1984), and using the estimated MMD between the potential outcomes as a test statistic. Throughout we will refer to $\widehat{\mathrm{MMD}}[Y(1), Y(0), \mathcal{H}_\mathcal{Y}]$ as the *distributional average treatment effect* (DATE), due to the similarity between this and the average treatment effect statistic in Section 2.1.

Analogously to the average treatment effect, the distributional average treatment effect is only identified from data under both-sided overlap. Therefore we now introduce a second target based of the effect of treatment on the treated, the *distributional effect of treatment on the treated* (DETT):

$$\mathrm{MMD}\left[Y(1)_{\{T=1\}}, Y(0)_{\{T=1\}}, \mathcal{H}_\mathcal{Y}\right] := \left\| \mu_{\{Y(1)|T=1\}} - \mu_{\{Y(0)|T=1\}} \right\|_{\mathcal{H}_\mathcal{Y}}.$$

Estimation this quantity requires estimation of the embeddings $\mu_{\{Y(0)|T=1\}}, \mu_{\{Y(1)|T=1\}}$. Since $P(Y(1) \mid T = 1) = P(Y \mid T = 1)$ we can estimate the latter embedding using observed samples of $Y$ for individuals with

$T = 1$. This means the only challenge is the estimation of $\mu_{\{Y(0)|T=1\}}$ which Muandet esimtate using the fact that:

$$\mu_{Y(0)|T=1} = \mathbb{E}_{P(X|T=1)} \left[ \mu_{Y|X,T=0} \right].$$

This means a finite sample estimate of the conditional mean embedding from Section 2.2 can be used to compute a finite sample estimate of the mean embedding.

Alternatively, Bellot & van der Schaar (2021) also target the distributional effect of treatment on the treated using the fact that:

$$\mu_{Y(0)|T=1} = \mathbb{E} \left[ \ell(Y, \cdot) \mathbb{1}\{T = 0\} w(X, 0) \right].$$

Both of these estimators lead to test statistics based on the distributional effect of treatment on the treated.

## 3 Doubly Robust Counterfactual Mean Embeddings

In this section we build on previous methodology by considering estimators for the causal mean embeddings that are built on doubly robust (DR) estimators of causal effects (Robins & Rotnitzky, 1995). We propose new estimators for the distributional average treatment effect and the distributional effect of treatment on the treated. We prove that both of these estimators have the doubly robust property and so they will converge to the true causal mean embedding if either of the two models underlying them converges to the true model. We call this set of approaches *Doubly Robust Counterfactual Mean Embeddings*.

### 3.1 Doubly robust estimator of the distributional average treatment effect (DATE)

Firstly, based on the doubly robust estimator of the average treatment effect we note that the embedding of the potential outcome, $Y(t)$, can also be written as:

$$\begin{aligned}
\mu_{Y(t)} &= \mathbb{E}[\ell(Y(t), \cdot) - \mu_{Y|X,T=t}] + \mathbb{E}[\mu_{Y|X,T=t}] \\
&= \mathbb{E}\left[ \frac{\mathbb{1}\{T = t\} \left( \ell(Y, \cdot) - \mu_{Y|X,T=t} \right)}{e(X, t)} \right] + \mathbb{E}[\mu_{Y|X,T=t}] \\
&= \mathbb{E}\left[ \frac{\mathbb{1}\{T = t\} \left( \ell(Y, \cdot) - \mu_{Y|X,T=t} \right)}{e(X, t)} + \mu_{Y|X,T=t} \right].
\end{aligned}$$

This is a kernelised version of the doubly robust estimator of the treatment mean (Robins & Rotnitzky, 1995); it can be estimated from finite samples by fitting models $\hat{e}(x, t), \hat{\mu}_{Y|X=x,T=t}$ on a training set and then averaging them over the test set as:

$$\hat{\mu}_{Y(t)}^{\mathrm{DR}} = \frac{1}{n} \sum_{i=1}^{n} \left\{ \frac{\mathbb{1}\{t_i = t\} \left( \ell(y_i, \cdot) - \hat{\mu}_{Y|X=x_i,T=t} \right)}{\hat{e}(x_i, t)} + \hat{\mu}_{Y|X=x_i,T=t} \right\}. \tag{2}$$

Take a constant train/test split and let $\hat{e}_n(X, t)$ and $\hat{\mu}_{Y|X,T=t}^{(n)}$ be the models that arise from a total sample of size $n$. Now under the assumption that the propensity and estimated propensity are uniformly bounded, the following theorem shows that $\hat{\mu}_{Y(t)}^{\mathrm{DR}}$ has the *doubly robust* property that it converges to the true embedding as long as either of the propensity model or the conditional mean embedding converges. Furthermore, it is also doubly rate robust, in the sense that it achieves a convergence rate that is the product of the convergence rates of these working models.

**Theorem 1.** *Assuming that $e(x, t)$ is strongly bounded away from zero, as well as additional overlap assumptions in Appendix A.1, we have that if the estimators $\hat{e}_n(X, t)$ and $\hat{\mu}_{Y|X,T=t}^{(n)}$ satisfy the following:*

$$\|\hat{e}_n(X, t) - e(X, t)\|_2 = \mathcal{O}_P(\gamma_{e,n}), \qquad \left\| \left\| \hat{\mu}_{Y|X,T=t}^{(n)} - \mu_{Y|X,T=t} \right\|_{\mathcal{H}_Y} \right\|_2 = \mathcal{O}_P(\gamma_{r,n})$$

*with $\gamma_{e,n} = O(1)$ and $\gamma_{r,n} = O(1)$. Then:*

$$\left\| \mu_{Y(t)} - \hat{\mu}_{Y(t)}^{\mathrm{DR}} \right\|_{\mathcal{H}_Y} = \mathcal{O}_P(\max\{n^{-\frac{1}{2}}, \gamma_{r,n} \gamma_{e,n}\}).$$

Therefore, if $\gamma_{r,n}\gamma_{e,n} = O(n^{-\frac{1}{2}})$ we obtain the parametric convergence rate.

These statistics can be used to form a doubly robust estimator of the distributional average treatment effect as:

$$\widehat{\mathrm{MMD}}_{\mathrm{DR}}[Y(1), Y(0), \mathcal{H}_{\mathcal{Y}}] = \left\| \hat{\mu}_{Y(1)}^{\mathrm{DR}} - \hat{\mu}_{Y(0)}^{\mathrm{DR}} \right\|_{\mathcal{H}_{\mathcal{Y}}}.$$

We derive the closed form of this statistic squared in Appendix B.1. This will converge to the correct MMD if both $\hat{\mu}_{Y(1)}^{\mathrm{DR}}$ and $\hat{\mu}_{Y(0)}^{\mathrm{DR}}$ converge to the true embedding at a rate that is a maximum of both their rates.

### 3.2 Doubly robust estimator of the Distributional Effect of Treatment on the Treated (DETT)

We now turn to estimating the distributional effect of treatment on the treated[2], again we form a kernel version of the standard doubly robust estimator of the distributional effect of treatment on the treated (Moodie et al., 2018). This leads from the fact that the mean embedding of the counterfactual distribution $P(Y(t) \mid T = t')$ can be written as:

$$\begin{aligned}
\mu_{Y(t)|T=t'} &= \mathbb{E}[\ell(Y(t), \cdot)] \\
&= \mathbb{E}[\ell(Y(t), \cdot) - \mu_{Y|X,T=t} \mid T = t'] + \mathbb{E}[\mu_{Y|X,T=t} \mid T = t'] \\
&= \mathbb{E}\left[ \frac{e(X, t')P(T = t)}{e(X, t)P(T = t')} \left( \ell(Y, \cdot) - \mu_{Y|X,T=t} \right) \mid T = t \right] + \mathbb{E}[\mu_{Y|X,T=t} \mid T = t'] \\
&= \mathbb{E}\left[ \frac{P(T = t)}{P(T = t')} w(X, t) \left( \ell(Y, \cdot) - \mu_{Y|X,T=t} \right) \mid T = t \right] + \mathbb{E}[\mu_{Y|X,T=t} \mid T = t'],
\end{aligned}$$

where $w(x, t) = \frac{1 - e(x,t)}{e(x,t)}$.

By fitting models $\hat{w}(x, t)$ and $\hat{\mu}_{Y|X=x,T=t}$ on training samples we get a finite sample estimate of the distributional effect of treatment on the treated as:

$$\hat{\mu}_{Y(t)|T=t'}^{\mathrm{DR}} = \frac{1}{n_{t'}} \sum_{i=1}^{n} \left( \mathbb{1}\{t_i = t\} \, \hat{w}(x_i, t) \left( \ell(y_i, \cdot) - \hat{\mu}_{Y|X=x_i,T=t} \right) + \mathbb{1}\{t_i = t'\} \, \hat{\mu}_{Y|X=x_i,T=t} \right)$$

where $n_t = \sum_i \mathbb{1}\{t_i = t\}$. Again letting $\hat{e}_n(x, t)$ and[3] $\hat{\mu}_{Y|X,T=t}^{(n)}$ be the respective models trained on a set of size $n$ we have that:

**Theorem 2.** *Under overlap assumptions given in the appendix, we have that if the estimators $\hat{e}_n(X, t)$ and $\hat{\mu}_{Y|X,T=t}^{(n)}$ satisfy the following:*

$$\left\| \hat{e}_n(X, t) - e(X, t) \right\|_2 = \mathcal{O}_P(\gamma_{e,n}), \qquad \left\| \left\| \hat{\mu}_{Y|X,T=t}^{(n)} - \mu_{Y|X,T=t} \right\|_{\mathcal{H}_Y} \right\|_2 = \mathcal{O}_P(\gamma_{r,n})$$

*where $\gamma_{e,n} = O(1)$ and $\gamma_{r,n} = O(1)$ then:*

$$\left\| \mu_{Y(t)|T=t'} - \hat{\mu}_{Y(t)|T=t'}^{\mathrm{DR}} \right\|_{\mathcal{H}_Y} = \mathcal{O}_P(\max\{n^{-\frac{1}{2}}, \gamma_{r,n}\gamma_{e,n}\}).$$

Applying this we can now estimate the relevant MMD for between treated and untreated for this population as:

$$\widehat{\mathrm{MMD}}_{\mathrm{DR}}\left[ Y(1)_{\{T=1\}}, Y(0)_{\{T=1\}}, \mathcal{H}_{\mathcal{Y}} \right] = \left\| \hat{\mu}_{Y(t)|T=t'}^{\mathrm{DR}} - \frac{1}{n_{t'}} \sum_{i=1}^{n} \mathbb{1}\{t_i = t'\} \, \ell(y_i, \cdot) \right\|_{\mathcal{H}_{\mathcal{Y}}}$$

Which has the same convergence rate as $\hat{\mu}_{Y(t)|T=t'}^{\mathrm{DR}}$. Therefore, we only need strict one-sided overlap for convergence, as $\hat{\mu}_{Y(t)|T=t'}^{\mathrm{DR}}$ just requires $e(x, t) > \epsilon$. Again we derive the squared statistic and give it in closed form in Appendix B.2.

---

[2] We leave the $t$ arbitrary, so if $t = 1$ this would form an effect of treatment on the control. For simplicity we do not distinguish the two cases.

[3] Any model for $w(x, t)$ leads to a model for $e(x, t)$ directly.

## 4 Permutation testing for Distributional Treatment Effects

We now apply both test statistics to the problem of testing for distributional treatment effects, taking a permutation based approach. As is standard in this context we test against Fisher's sharp null (Fisher, 1936; Rosenbaum, 2002)[4]:

$$H_0 : Y_i(1) = Y_i(0).$$

In order to apply a permutation algorithm, we must choose permutations such that the treatment vector is *exchangeable* under them. That is, if we let $n$ be the size of our dataset $\mathcal{D} = (\mathbf{X}, \mathbf{Y}, \mathbf{T})$ and $\mathrm{Sym}_n$ be the permutation group of size $n$ we want to restrict to permutations, $\sigma \in \mathrm{Sym}_n$, such that:

$$(\mathbf{X}, \mathbf{Y}, \mathbf{T}) \overset{d}{=} (\mathbf{X}, \mathbf{Y}, \mathbf{T}_\sigma),$$

where $\mathbf{T}_\sigma = \left(T_{\sigma(i)}\right)_{i=1}^n$ is the permuted treatment vector.

A standard way to ensure that the permutations have this property is to through matching (Stuart, 2010), where we split the data into matched sets, such that within each set we have one treated individual. These matched sets are formed using some measure of 'similarity' of individuals, such as the Mahalanobis distance on covariates, or through the propensity score. Rosenbaum (2002) argues that if this matching is exact, so that there are no differences in propensity between matched individuals, then the treatment is exchangeable under permutations that preserve the matched sets. That is if we let $M \subset \mathrm{Sym}_n$ be the subset of permutations that only permute within matched sets and sample some $\sigma \in M$, then $\mathbf{T}$ is exchangeable under $\sigma$. Therefore if we fix a statistic $S$, in our case either the DATE or DETT statistic, and randomly sample $\sigma_1, \ldots, \sigma_m$ from $M$, we have that:

$$S(\mathbf{X}, \mathbf{Y}, \mathbf{T}), S(\mathbf{X}, \mathbf{Y}, \mathbf{T}_{\sigma_1}), \ldots, S(\mathbf{X}, \mathbf{Y}, \mathbf{T}_{\sigma_m}) \text{ are exchangeable.}$$

This allows means that we can define the p-value of the test as:

$$p = \frac{1 + \sum_{i=1}^m \mathbb{1}\{S(\mathbf{X}, \mathbf{Y}, \mathbf{T}_{\sigma_i}) \geq S(\mathbf{X}, \mathbf{Y}, \mathbf{T})\}}{m + 1},$$

where this is now a valid p-value in the sense that $P(p \leq \alpha) \leq \alpha$ under the null. However, permuting in this way creates computational challenges when $S$ is one of the proposed statistics. Namely, the calculation of these statistics requires the fitting of a propensity model and conditional mean embedding, and this would need to be repeated for every permutation.

To resolve this we first form the matched sets, and then randomly split into train/test sets, $\mathcal{D}_{\mathrm{Tr}}, \mathcal{D}_{\mathrm{Te}}$, such that each matching set is fully contained in one of the datasets. Now let $M_{|\mathcal{D}_{\mathrm{Tr}}|}$ be the set of permutations on $\mathcal{D}$ which leave $\mathcal{D}_{\mathrm{Te}}$ fixed and preserve the matching on $\mathcal{D}_{\mathrm{Tr}}$, and vice versa for $\mathcal{D}_{\mathrm{Te}}$. We then sample $N$ random permutations $\{\sigma_1, \ldots, \sigma_N\}$ from $M_{|\mathcal{D}_{\mathrm{Tr}}|}$, and form the set $\tilde{M}$ of permutations as:

$$\tilde{M}_N = \{\sigma \circ \pi : \sigma \in \{Id\} \cup \{\sigma_1, \ldots, \sigma_N\}, \pi \in M_{\mathcal{D}_{\mathrm{Te}}}\},$$

Then the following demonstrates by applying results from Ramdas et al. (2023) that we can form a valid permutation test by randomly sampling permutations from $\tilde{M}$:

**Proposition 1.** *Under exact matching, if we sample $\tau_0, \ldots, \tau_m$ from $\tilde{M}_N$ we have that the following is a valid p-value:*

$$p = \frac{1 + \sum_{i=1}^m \mathbb{1}\{S(\mathbf{X}, \mathbf{Y}, \mathbf{T}_{\tau_i}) \geq S(\mathbf{X}, \mathbf{Y}, \mathbf{T})\}}{m + 1}$$

*for any statistic $S$ and number of sampled permutations $N$.*

---

[4]We note that while the test must be formulated against Fisher's sharp null, it will generally have power against alternatives which show distributional causal effects, so $Y(1) \overset{d}{\neq} Y(0)$.

This alleviates the computational challenges associated with using the DATE or DETT statistics as there is now a controllable parameter $N$ which determines how many models must be trained. Increasing $N$ will mean training more models and so will decrease the variance in $p$, however it will incur greater computational cost.

To our knowledge, this is the first time this permutation strategy has appeared in the literature. Moreover, this procedure could be applied to efficiently use other trainable test statistics within permutation testing, such as standard doubly robust estimators of the causal mean or more general permutation tests for assessing the performance of predictive models as in Ojala & Garriga (2010).

The result in Proposition 1 relies on exact matching, which is a common assumption in proving the validity of conditional permutation tests Rosenbaum (2002). However recent results in the area have shown that p-values arising from matched permutations are approximately valid under inexact matching (Berrett et al., 2020; Pimentel, 2022). We expect and empirically observe that similar results hold in our case.

## 5 Experiments

### 5.1 Fit tests for Doubly Robust Counterfactual Mean Embeddings

To demonstrate the convergence properties implied by our theoretical results in Section 3 we fit our statistics on simulated data from the following data generating processes:

$$X \sim \mathcal{N}(0, I_9) \qquad p = \left(\frac{1}{1 + \exp\left(\boldsymbol{a}^\top X\right)}\right)^2 - \mathbb{E}_X\left[\left(\frac{1}{1 + \exp\left(\boldsymbol{a}^\top X\right)}\right)^2\right]$$

$$T \sim \mathrm{Ber}(p) \qquad Y = \boldsymbol{b}^\top X + \beta T + \epsilon,$$

where $\epsilon \sim \mathcal{N}(0, \sigma^2)$ and the values of $\boldsymbol{a}, \boldsymbol{b} \in \mathbb{R}^9$ and $\beta, \sigma \in \mathbb{R}$ are given in Appendix C. This is similar to the setting in Muandet et al. (2021). We also simulate from:

$$T \sim \mathrm{Ber} \qquad X \sim \mathcal{N}(0, (1 + \alpha T)I_{10}) \qquad Y = f_T(X) + \epsilon' \qquad \epsilon' \sim \mathcal{N}(0, (\sigma')^2)$$

where $f_0, f_1 : \mathbb{R}^{10} \to \mathbb{R}$ and $\alpha, \sigma'$ are given in the Appendix. The setting here is the same as in Bellot & van der Schaar (2021). For both settings we fit a linear logistic regression for the propensity score so that the model is incorrectly specified. In first data generating process we have both-sided overlap in the true propensity and so plots (a) and (b) in Figure 2 demonstrate that the doubly robust embeddings converge for both interventional values. Due to an incorrectly specified propensity model the counterfactual mean embeddings of Muandet et al. (2021) do not converge.

For the second data generating process, increasing the value of $b$ creates a setting where the overlap is more one-sided. As such plots (c) and (d) in Figure 2 demonstrate that only $\hat{\mu}^{\mathrm{DR}}_{Y(1)}$ and $\hat{\mu}^{\mathrm{DR}}_{Y(1)|T=0}$ converge quickly to the true embeddings.

Plots of the true propensity for both simulations can be found in Appendix C.1.

### 5.2 Testing For Distributional Effects on Simulated Data

We now apply these statistics to the testing of distributional causal effects where the data is simulated from:

$$X \sim \mathcal{N}(0, I_9), \quad p = \left(\frac{1}{1 + \exp\left(\boldsymbol{a}^\top X\right)}\right), \tag{3}$$

$$T \sim \mathrm{Ber}(p), \quad Y = \boldsymbol{b}^\top X + \beta(2Z - 1)T + \epsilon, \quad \epsilon \sim \mathcal{N}(0, \sigma^2) \tag{4}$$

Where $\alpha, \beta \in \mathbb{R}^9$ and $\sigma \in \mathbb{R}$ are as in Section 5.1 and the variable $Z$ is 1, a $\mathrm{Ber}(\frac{1}{2})$ or $\mathrm{Unif}([0, 1])$. The second two settings capture when there is a distributional causal effect which does not shift the causal means.

In Figure 3 we plot the rejection rate of the tests based on four statistics, the DATE and DETT statistics estimated without conditional mean embeddings, and the DATE and DETT statistics estimated with conditional mean embeddings denoted by DR-DATE and DR-DETT respectively. The matching for all statistics is done via logistic regression and we apply the permutation from Section 4. We compare our methods against Double Machine Learning (Chernozhukov et al., 2018) and Targeted Maximum Likelihood

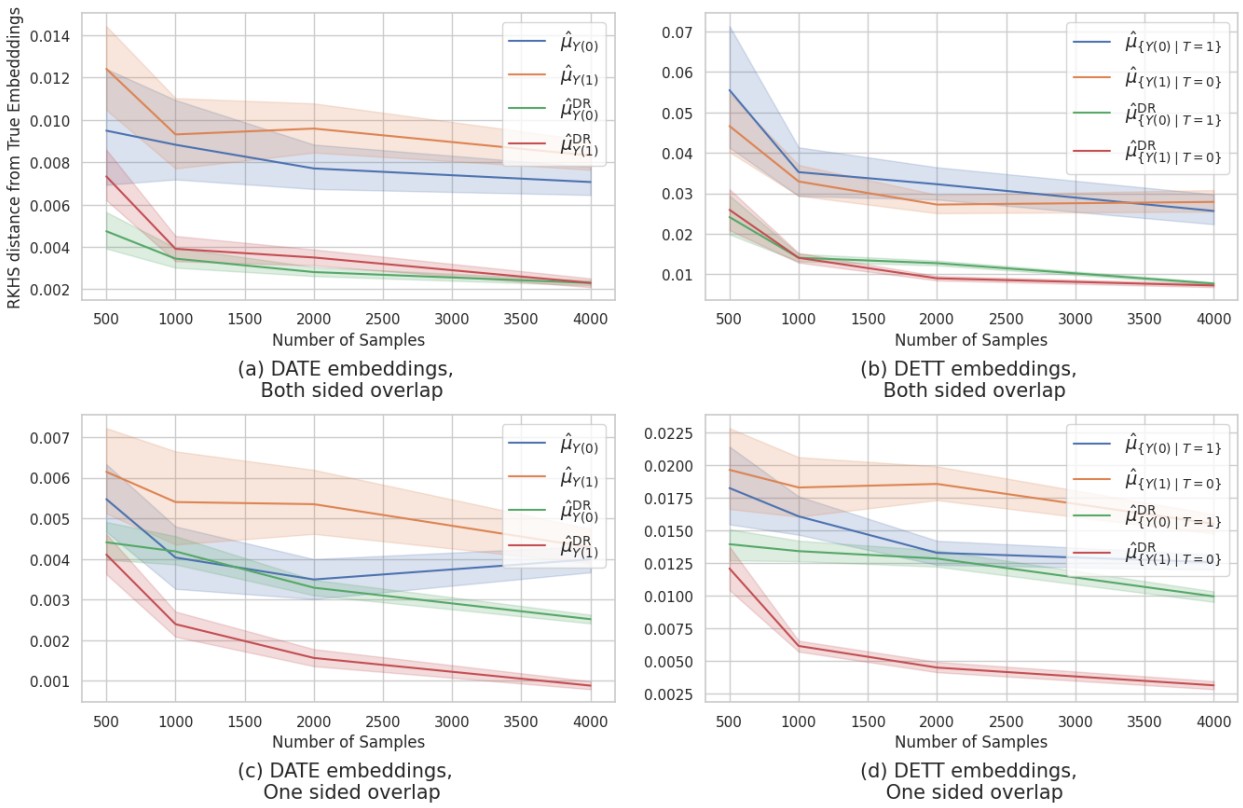

Figure 2: These plots demonstrate the convergence of the doubly robust estimators when the propensity model is misspecified. The left plots show the embeddings required for DATE, whilst the right plots show those needed for DETT. We vary between two data generating process given in Section 5.1, aiming to show both sided and one sided overlap of the propensity score.

Estimation (Van Der Laan & Rubin, 2006) as baselines to effectively pick up a shift in causal means. We run these experiments with 2000 data points, rejecting at the 0.05 significance level. Power plots showing the rejection rates over 100 re-runs of this experiment are displayed in Figure 3.

In this experiment we observe similar performance for all distributional test statistics. We believe this is due to the fact that both the true and estimated propensities are linear functions of the covariates, and so all statistics will correctly fit the causal mean embeddings. Plot (a) demonstrates that when the shift is in the mean, as expected our methods will perform worse than the baselines. This is due to the cost of targeting distributional effects over simply mean shifts. However, plots (b) and (c) demonstrate that when the causal effect is only distributional, our statistics can correctly reject the null, unlike these traditional methods.

## 5.3 Semi-Synthetic and Real World Data

Finally, to evaluate the performance of our tests within more realistic settings we evaluate on a selection of semi-synthetic data. We evaluate on two standard semi-synthetic tasks, the infant health and development program (IDHP) introduced in Hill (2011), the linked births and deaths data (LBIDD) (Shimoni et al., 2018).

As both of these datasets are semi-synthetic, we have access to the counterfactuals and so can simulate realistic data under the null hypothesis. In order to prevent problems with extreme propensity scores we remove any data for which $e(x, t) < 0.03$ for any value of $t$. We again use logistic regression matching and weights model.

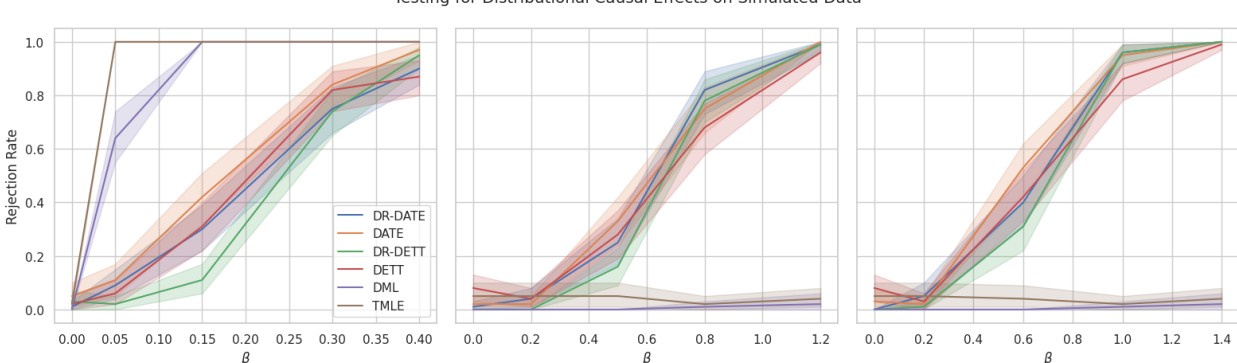

Figure 3: Simulated tests for distributional causal effects where the $\beta$ parameter controls the size of the effect. In plot $(a)$, $\beta$ introduces a shift in the mean only, whereas in $(b)$ and $(c)$ the shift only affects higher moments of the distribution. All simulations with $n = 2000$ samples.

Table 1: Rejection rates at the 0.05 level on semi-synthetic datasets.

| Dataset | Hypothesis | DATE | DR-DATE | DETT | DR-DETT | DML |
|---|---|---|---|---|---|---|
| IDHP $(n = 747)$ | $H_0$ | $0.02 \pm 0.03$ | $0.05 \pm 0.04$ | $0.01 \pm 0.02$ | $0.03 \pm 0.03$ | $0.04 \pm 0.04$ |
| | $H_1$ | $1.00 \pm 0$ | $1.00 \pm 0$ | $1.00 \pm 0$ | $0.99 \pm 0.03$ | $1.00 \pm 0$ |
| LBIDD $(n = 1000)$ | $H_0$ | $0.02 \pm 0.03$ | $0.03 \pm 0.03$ | $0.00 \pm 0$ | $0.00 \pm 0$ | $0.17 \pm 0.10$ |
| | $H_1$ | $0.85 \pm 0.07$ | $0.82 \pm 0.07$ | $0.13 \pm 0.06$ | $0.08 \pm 0.06$ | $1.00 \pm 0$ |
| LBIDD $(n = 2500)$ | $H_0$ | $0.03 \pm 0.03$ | $0.03 \pm 0.03$ | $0.00 \pm 0$ | $0.00 \pm 0$ | $0.00 \pm 0$ |
| | $H_1$ | $0.82 \pm 0.08$ | $0.75 \pm 0.09$ | $0.19 \pm 0.08$ | $0.13 \pm 0.06$ | $1.00 \pm 0$ |

The results can be found in Table 1, which shows all kernel based test statistics produce valid p-values under the null hypothesis. Further the DATE based test statistics demonstrate strong power against the alternative for all three datasets. The DETT test statistic on the other hand only shows a good level of power on the IDHP dataset, with more samples required to reject on the LBIDD dataset.

## 6  Conclusion

We have proposed new doubly robust estimators for the kernel mean embedding of causal distributions. Our theoretical experimental results show that these converge in a wider array of circumstances than the original estimators of these quantities. Further, we applied these embeddings to the problem of testing for distributional causal effects. We used a permutation based approach for testing, and proposed a permutation that allows us to use our doubly robust statistics within permutation testing. We prove the validity of this permutation under exact matching and experimentally validate this approach under inexact matching. The results show that our test statistics are able to pick up causal effects that only manifest as distributional shifts, where traditional mean shift methods fail.

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

# A Proofs

For ease of writing, throughout this appendix we will let:

$$r(x,t) := \mu_{Y|X=x,T=t} \tag{5}$$

$$\hat{r}_n(x,t) := \hat{\mu}^{(n)}_{Y|X=x,T=t} \tag{6}$$

## A.1 Proof of Theorem 1

This proof follows under the following standard assumptions[5]:

- The propensity is uniformly bounded away from zero. So there exist an $\epsilon > 0$ such that $e(x,t) > \epsilon$ for all $x$ in the support of the covariates.

- We assume that the estimated propensity score is uniformly bounded away from 0 and 1 on the support of $P(X)$. So for all $x$ in the support of $P(X)$ and for all $n$ we have $\delta < \hat{e}_n(x,t)$ for some $\delta > 0$.

Now let $m$ be the size of the test set we average over. So $m = \mathcal{O}(n)$ where $n$ is the sample size.

First note due to the consistency property we can write $k(y_i, \cdot)\mathbb{1}\{t_i = t\} = k(y_i(t), \cdot)\mathbb{1}\{t_i = t\}$. This allows us to rewrite the DR estimator as:

$$\hat{\mu}^{\mathrm{DR}}_{Y(t)} = \frac{1}{m} \sum_{i=1}^{m} \left\{ \ell(y_i(t), \cdot) + \frac{(\mathbb{1}\{t_i = 1\} - \hat{e}_n(x_i,t))\,(\ell(y_i(t), \cdot) - \hat{r}_n(x_i,t))}{\hat{e}_n(x_i,t)} \right\}.$$

Now letting:

$$\epsilon^{\mathrm{DR}}_{Y(t)} = \frac{1}{m} \sum_{i=1}^{m} \frac{(\mathbb{1}\{t_i = t\} - \hat{e}_n(x_i,t))\,(\ell(y_i(t), \cdot) - \hat{r}_n(x_i,t))}{\hat{e}_n(x_i,t)},$$

we can write:

$$\left\| \mu_{Y(t)} - \hat{\mu}^{\mathrm{DR}}_{Y(t)} \right\|_{\mathcal{H}_Y} \leq \left\| \mu_{Y(t)} - \frac{1}{m} \sum_{i=1}^{m} \ell(y_i(t), \cdot) \right\|_{\mathcal{H}_Y} + \left\| \epsilon^{\mathrm{DR}}_{Y(t)} \right\|_{\mathcal{H}_Y}.$$

As $\frac{1}{m} \sum_{i=1}^{m} \ell(y_i(t), \cdot)$ is an unbiased estimator for $\mu_{Y(t)}$ with parametric convergence rate $\mathcal{O}(m^{-\frac{1}{2}}) = \mathcal{O}(n^{-\frac{1}{2}})$, the convergence rate of our estimator is entirely determined by the convergence rate of $\left\| \epsilon^{\mathrm{DR}}_{Y(t)} \right\|_{\mathcal{H}_Y}$ so consider:

$$\mathbb{E} \left\| \epsilon^{\mathrm{DR}}_{Y(t)} \right\|^2_{\mathcal{H}_Y}$$

$$= \mathbb{E} \left[ \frac{1}{m^2} \sum_{i,j=1}^{m} \frac{(\mathbb{1}\{t_i = t\} - \hat{e}_n(x_i,t))\,(\mathbb{1}\{t_j = t\} - \hat{e}_n(x_j,t))\,\langle y_j(t) - \hat{r}_n(x_j,t), y_i(t) - \hat{r}_n(x_i,t)\rangle_{\mathcal{H}_Y}}{\hat{e}_n(x_i,t)\hat{e}_n(x_j,t)} \right]$$

$$= \mathbb{E} \left[ \frac{1}{m^2} \sum_{i \neq j} \frac{(\mathbb{1}\{t_i = t\} - \hat{e}_n(x_i,t))\,(\mathbb{1}\{t_j = t\} - \hat{e}_n(x_j,t))\,(\langle y_j(t) - \hat{r}_n(x_j,t), y_i(t) - \hat{r}_n(x_i,t)\rangle_{\mathcal{H}_Y})}{\hat{e}_n(x_i,t)\hat{e}_n(x_j,t)} \right. \tag{7}$$

$$\left. + \frac{1}{m^2} \sum_{i=j} \left\{ \frac{(\mathbb{1}\{t_i = t\} - \hat{e}_n(x_i,t))\,\|y_i(t) - \hat{r}_n(x_i,t)\|_{\mathcal{H}_Y}}{\hat{e}_n(x_i,t)} \right\}^2 \right] \tag{8}$$

Now let $(X, T, Y(t))$ be a random sample. We may write the term (8) as:

$$\frac{1}{m} \mathbb{E} \left\{ \frac{(\mathbb{1}\{T = t\} - \hat{e}_n(X,t))\,\|Y(t) - \hat{r}_n(X,t)\|_{\mathcal{H}_Y}}{\hat{e}_n(X,t)} \right\}^2.$$

---

[5]These are the same as in for example Kennedy et al. (2017)

By applying bounded propensity and bounded kernels we have that this expectation is bounded, which gives the term is bounded by $C_1/n$. Now for the first term let $(\tilde{X}, \tilde{T}, \tilde{Y}(t))$ be a second independent sample. We may write (7) as:

$$
\frac{m-1}{m}\mathbb{E}\left[\frac{\left(\mathbb{1}\{T=t\}-\hat{e}_n(X,t)\right)\left(\mathbb{1}\{\tilde{T}=t\}-\hat{e}_n(\tilde{X},t)\right)\left\langle Y(t)-\hat{r}_n(X,t),\tilde{Y}(t)-\hat{r}_n(\tilde{X},t)\right\rangle_{\mathcal{H}_Y}}{\hat{e}_n(X,t)\hat{e}_n(\tilde{X},t)}\right]
$$

$$
=\frac{m-1}{m}\mathbb{E}\left[\mathbb{E}\left[\frac{\left(\mathbb{1}\{T=t\}-\hat{e}_n(X,t)\right)\left(\mathbb{1}\{\tilde{T}=t\}-\hat{e}_n(\tilde{X},t)\right)\left\langle Y(t)-\hat{r}_n(X,t),\tilde{Y}(t)-\hat{r}_n(\tilde{X},t)\right\rangle_{\mathcal{H}_Y}}{\hat{e}_n(X,t)\hat{e}_n(\tilde{X},t)}\Bigg|X,\tilde{X}\right]\right]
$$

$$
=\frac{m-1}{m}\mathbb{E}\left[\mathbb{E}\left[\frac{\left(\mathbb{1}\{T=t\}-\hat{e}_n(X,t)\right)\left(\mathbb{1}\{\tilde{T}=t\}-\hat{e}_n(\tilde{X},t)\right)}{\hat{e}_n(X,t)\hat{e}_n(\tilde{X},t)}\Bigg|X,\tilde{X}\right]\mathbb{E}\left[\left\langle Y(t)-\hat{r}_n(X,t),\tilde{Y}(t)-\hat{r}_n(\tilde{X},t)\right\rangle_{\mathcal{H}_Y}\Bigg|X,\tilde{X}\right]\right]
$$

$$
=\frac{m-1}{m}\mathbb{E}\left[\frac{\left(e(X,t)-\hat{e}_n(X,t)\right)\left(e(\tilde{X},t)-\hat{e}_n(\tilde{X},t)\right)}{\hat{e}_n(X,t)\hat{e}_n(\tilde{X},t)}\left\langle r(X,t)-\hat{r}_n(X,t),r(\tilde{X},t)-\hat{r}_n(\tilde{X},t)\right\rangle_{\mathcal{H}_Y}\right]
$$

$$
\leq\frac{m-1}{m}\left(\mathbb{E}\left[\left(\frac{\left(e(X,t)-\hat{e}_n(X,t)\right)\left(e(\tilde{X},t)-\hat{e}_n(\tilde{X},t)\right)}{\hat{e}_n(X,t)\hat{e}_n(\tilde{X},t)}\right)^2\right]\mathbb{E}\left[\left\langle r(X,t)-\hat{r}_n(X,t),r(\tilde{X},t)-\hat{r}_n(\tilde{X},t)\right\rangle_{\mathcal{H}_Y}^2\right]\right)^{\frac{1}{2}}
$$

$$
\leq\frac{m-1}{m}\mathbb{E}\left[\left(\frac{e(X,t)-\hat{e}_n(X,t)}{\hat{e}_n(X,t)}\right)^2\right]\mathbb{E}\left[\|r(X,t)-\hat{r}_n(X,t)\|_{\mathcal{H}_Y}^2\right]
$$

$$
\leq C_2\mathbb{E}\left[(e(X,t)-\hat{e}_n(X,t))^2\right]\mathbb{E}\left[\|r(X,t)-\hat{r}_n(X,t)\|_{\mathcal{H}_Y}^2\right].
$$

Therefore we have:

$$
\mathbb{E}\left\|\epsilon_{Y(t)}^{\mathrm{DR}}\right\|_{\mathcal{H}_Y}^2 \leq \frac{C_1}{m} + C_2\mathbb{E}\left[(e(X,t)-\hat{e}_n(X,t))^2\right]\mathbb{E}\left[\|r(X,t)-\hat{r}_n(X,t)\|_{\mathcal{H}_Y}^2\right].
$$

This gives that the rate of convergence is controlled by $\mathbb{E}\left(e(X,t)-\hat{e}_n(X,t)\right)^2$ and $\mathbb{E}\|Y(t)-\hat{r}_n(X,t)\|_{\mathcal{H}_Y}^2$. By applying Jensen's inequality we have $\mathbb{E}\left\|\epsilon_{Y(t)}^{\mathrm{DR}}\right\|_{\mathcal{H}_Y} = \mathcal{O}(\gamma_{r,n}\gamma_{e,n})$ and so by Markov's inequality $\left\|\epsilon_{Y(t)}^{\mathrm{DR}}\right\|_{\mathcal{H}_Y} = \mathcal{O}_P(\gamma_{r,n}\gamma_{e,n})$.

## A.2   Proof of Theorem 2

The proof follows a similar structure to that of Theorem 1 where we now have:

$$
\left\|\mu_{Y(t)|t=t'} - \hat{\mu}_{Y(t)|t=t'}^{\mathrm{DR}}\right\|_{\mathcal{H}_Y} \leq \left\|\mu_{Y(t)} - \frac{1}{n_{t'}}\sum_{i=1}^n \mathbb{1}\{t_i=t'\}\,\ell(y_i(t),\cdot)\right\|_{\mathcal{H}_Y}
$$
$$
+ \left\|\left(\frac{1}{n_{t'}}\sum_{i=1}^n \mathbb{1}\{t_i=t'\}\,\ell(y_i(t),\cdot)\right) - \hat{\mu}_{Y(t)|t=t'}^{\mathrm{DR}}\right\|.
$$

Now we have:

$$
\left( \frac{1}{n_{t'}} \sum_{i=1}^{n} \mathbb{1}\{t_i = t'\} \ell(y_i(t), \cdot) \right) - \hat{\mu}_{Y(t)|t=t'}^{\mathrm{DR}}
$$

$$
= \frac{1}{n_{t'}} \sum_{i=1}^{n} \left\{ \mathbb{1}\left\{ t_i = t' \right\} \ell(y_i(t), \cdot) - \left( \mathbb{1}\{t_i = t\} \, w(x_i, t) \left( \ell(y_i(t), \cdot) - r(x_i, t) \right) + \mathbb{1}\{t_i = t'\} r(x_i, t) \right) \right\}
$$

$$
= -\frac{1}{n_{t'}} \sum_{i=1}^{n} \left\{ \mathbb{1}\{t_i = t\} \, w(x_i, t) \left( \ell(y_i(t), \cdot) - r(x_i, t) \right) + \mathbb{1}\left\{ t_i = t' \right\} \left( r(x_i, t) - \ell(y_i(t), \cdot) \right) \right\}
$$

$$
= -\frac{1}{n_{t'}} \sum_{i=1}^{n} \left\{ \left( \ell(y_i(t), \cdot) - r(x_i, t) \right) \left( \mathbb{1}\{t_i = t\} \, w(x_i, t) - \mathbb{1}\left\{ t_i = t' \right\} \right) \right\}
$$

$$
= -\frac{1}{n_{t'}} \sum_{i=1}^{n} \left\{ \left( \ell(y_i(t), \cdot) - r(x_i, t) \right) \left( \mathbb{1}\{t_i = t\} \, w(x_i, t) - \mathbb{1}\{t_i = t'\} \right) \right\}
$$

$$
= \frac{1}{n_{t'}} \sum_{i=1}^{n} \left\{ \left( \ell(y_i(t), \cdot) - r(x_i, t) \right) \left[ \mathbb{1}\{t_i = t\} \left( \frac{1 - e(x_i, t)}{e(x_i, t)} \right) - \mathbb{1}\{t_i = t'\} \frac{e(x_i, t)}{e(x_i, t)} \right] \right\}
$$

$$
= \frac{1}{n_{t'}} \sum_{i=1}^{n} \left\{ \left( \ell(y_i(t), \cdot) - r(x_i, t) \right) \frac{\mathbb{1}\{t_i = t\} - \mathbb{1}\{t_i = t\} \, e(x_i, t) - \mathbb{1}\{t_i = t'\} \, e(x_i, t)}{e(x_i, t)} \right\}
$$

$$
= \frac{1}{n_{t'}} \sum_{i=1}^{n} \left\{ \left( \ell(y_i(t), \cdot) - r(x_i, t) \right) \frac{\mathbb{1}\{t_i = t\} - e(x_i, t)}{e(x_i, t)} \right\}
$$

$$
= \frac{n_{t'}}{n} \epsilon_{Y(t)}^{\mathrm{DR}}
$$

Now as $n_{t'}/n$ converges to a constant at rate $n^{-\frac{1}{2}}$ the convergence rate just depends on the rate that $\epsilon_{Y(t)}^{\mathrm{DR}}$ tends to zero. Therefore the analysis in the previous proof establishes the rate.

### A.3 Proof of Proposition 1

Ramdas et al. (2023) demonstrate that we can test the null hypothesis

$$H_0 : X_1, \ldots, X_n \text{ are exchangeable}$$

from a fixed set of permutations $S \subset \mathrm{Sym}_n$ (not necessarily a group) using a statistic $T$ by randomly sampling permutations $\sigma_0, \ldots, \sigma_M$ from $S$ uniformly and computing:

$$
p = \frac{1 + \sum_{i=1}^{M} \mathbb{1}\left\{ T(X_{\sigma_i \circ \sigma_0^{-1}}) \geq T(X) \right\}}{M + 1}, \tag{9}
$$

where $X_\sigma = \left( X_{\sigma(1)}, \ldots, X_{\sigma(n)} \right)$ is the permuted data. Ramdas et al. show that $p$ is a valid p-value in the sense that $P(p \leq \alpha) \leq \alpha$ under the null hypothesis.

In our method the data is placed into bins, denoted by $B_i$, where $i$ ranges over the total number of bins $k$, and under the assumption of exact matching the null hypothesis is:

$$H_0 : T_{B_{i_1}}, \ldots, T_{B_{i_n}} \text{ are exchangeable for each } i.$$

Further we have split the bins into training a $\mathcal{B}_{\mathrm{Tr}} = \{B_1, \ldots, B_m\}$ and $\mathcal{B}_{\mathrm{Te}} = \{B_{m+1}, \ldots, B_k\}$. For $B_i \in \mathcal{B}_{\mathrm{Te}}$ we set the possible permutations to be $S_{|B_i|}$ i.e. the full set of permutations on $B_i$. For each set $B_j \in \mathcal{B}_{\mathrm{Te}}$ we sample $\sigma_{1j}, \ldots, \sigma_{N+1}$ from $S_{|B_j|}$ and set $S_j = \{\sigma_{1j}, \ldots, \sigma_{N+1}\}$. We then set the total set of permutations $S$ to be:

$$
S = \left( \underset{j=1}{\overset{m}{\bigtimes}} S_j \right) \times \left( \underset{j=m+1}{\overset{k}{\bigtimes}} S_{|B_i|} \right)
$$

As the samples in each bin are exchangeable, and samples in distinct bins are independent, we have that our data is exchangeable under the permutations in $S$. Therefore we can sample from $S$ as in (9) to produce a valid p-value under $H_0$.

To go from this to the form in the proof note that sampling the random permutations for $S_j$, choosing a random permutation $\sigma_0$ from $S_j$ and then taking $\sigma_i \circ \sigma_0^{-1}$ is equivalent to sampling $N$ random permutations, $\sigma_{1j}, \ldots, \sigma_N$ and then sampling a random permutation from $\tilde{S} = \{\mathrm{Id}, \sigma_{1j}, \ldots, \sigma_N\}$.

# B  Derivation of Test Statistics

## B.1  Derivation of $\widehat{\mathrm{MMD}}^2_{\mathrm{DR}}[Y(1), Y(0), \mathcal{H}_{\mathcal{Y}}]$

To derive $(\widehat{\mathrm{MMD}}_{\mathrm{DR}}[Y(1), Y(0), \mathcal{H}_{\mathcal{Y}}])^2$ we let $e(x) = e(x, 1) = P(T = 1 \mid X = x)$. We now write:

$$
\begin{aligned}
\hat{\mu}^{\mathrm{DR}}_{Y(1)} - \hat{\mu}^{\mathrm{DR}}_{Y(0)} &= \frac{1}{m} \sum_{i=1}^{m} \left\{ \frac{t_i \left( \ell(y_i, \cdot) - \hat{r}(x_i, 1) \right)}{\hat{e}(x_i)} + \hat{r}(x_i, 1) - \left( \frac{(1 - t_i) \left( \ell(y_i, \cdot) - \hat{r}(x_i, 0) \right)}{1 - \hat{e}(x_i)} + \hat{r}(x_i, 0) \right) \right\} \\
&= \frac{1}{m} \sum_{i=1}^{m} \frac{t_i - e(x_i)}{e(x_i)(1 - e(x_i))} \left( \ell(y_i, \cdot) - (1 - e(x_i))\hat{r}(x_i, 1) + e(x_i)\hat{r}(x_i, 0) \right).
\end{aligned}
$$

Now we have $\hat{r}(x_i, 1) = \mathbf{k}_1^\top(x_i)\mathbf{W}_1\boldsymbol{\ell}_1$ and $\hat{r}(x_i, 0) = \mathbf{k}_0^\top(x_i)\mathbf{W}_0\boldsymbol{\ell}_0$. Further for ease of writing we let:

$$
\begin{aligned}
\tilde{\mathbf{k}}_0^\top(x_i) &= e(x_i)\mathbf{k}_0^\top(x_i), \\
\tilde{\mathbf{k}}_1^\top(x_i) &= (1 - e(x_i))\mathbf{k}_1^\top(x_i), \\
\alpha(x_i) &= \frac{(t_i - e(x_i))}{(e(x_i)(1 - e(x_i)))}.
\end{aligned}
$$

This allows us to write the above as

$$
= \frac{1}{m} \sum_{i=1}^{m} \alpha(x_i) \left( \ell(y_i, \cdot) + \tilde{\mathbf{k}}_0^\top(x_i)\mathbf{W}_0\boldsymbol{\ell}_0 - \tilde{\mathbf{k}}_1^\top(x_i)\mathbf{W}_1\boldsymbol{\ell}_1 \right),
$$

and plugging this in we obtain

$$
\begin{aligned}
(\widehat{\mathrm{MMD}}_{\mathrm{DR}}[Y(1), Y(0), \mathcal{H}_{\mathcal{Y}}])^2 &:= \left\| \hat{\mu}^{\mathrm{DR}}_{Y(1)} - \hat{\mu}^{\mathrm{DR}}_{Y(0)} \right\|^2_{\mathcal{H}_{\mathcal{Y}}} \\
&= \sum_{i,j} \alpha(x_i)\alpha(x_j) \left\langle \ell(y_i, \cdot) + \tilde{\mathbf{k}}_0^\top(x_i)\mathbf{W}_0\boldsymbol{\ell}_0 - \tilde{\mathbf{k}}_1^\top(x_i)\mathbf{W}_1\boldsymbol{\ell}_1, \ell(y_j, \cdot) + \tilde{\mathbf{k}}_0^\top(x_j)\mathbf{W}_0\boldsymbol{\ell}_0 - \tilde{\mathbf{k}}_1^\top(x_j)\mathbf{W}_1\boldsymbol{\ell}_1 \right\rangle \\
&= \sum_{i,j} \alpha(x_i)\alpha(x_j) \left( \ell(y_i, y_j) + \tilde{\mathbf{k}}_0^\top(x_i)\mathbf{W}_0\boldsymbol{\ell}_0(y_j) + \tilde{\mathbf{k}}_0^\top(x_j)\mathbf{W}_0\boldsymbol{\ell}_0(y_i) - \tilde{\mathbf{k}}_1^\top(x_i)\mathbf{W}_1\boldsymbol{\ell}_1(y_j) - \tilde{\mathbf{k}}_1^\top(x_j)\mathbf{W}_1\boldsymbol{\ell}_1(y_i) \right) \\
&+ \sum_{i,j} \alpha(x_i)\alpha(x_j) \left( \tilde{\mathbf{k}}_0^\top(x_i)\mathbf{W}_0\mathbf{L}_0\mathbf{W}_0\tilde{\mathbf{k}}_0(x_j) + \tilde{\mathbf{k}}_1^\top(x_i)\mathbf{W}_1\mathbf{L}_1\mathbf{W}_1\tilde{\mathbf{k}}_1(x_j) - \tilde{\mathbf{k}}_0^\top(x_i)\mathbf{W}_0\mathbf{L}_{0,1}\mathbf{W}_1\tilde{\mathbf{k}}_1(x_j) - \tilde{\mathbf{k}}_1^\top(x_i)\mathbf{W}_1\mathbf{L}_{0,1}\mathbf{W}_0\tilde{\mathbf{k}}_0(x_j) \right).
\end{aligned}
$$

Now we introduce some new notation: for a variable we will use a subscript to denote the value of $T$ we condition and a superscript to denote if the variable is in training or test. So $X_0^{\mathrm{Te}}$ is the test points with $T = 0$. Further let $K(X, \tilde{X}) = (k(x_i, \tilde{x}_j))_{i,j=1}^{i=n_X, j=n_{\tilde{X}}}$, so the kernel matrix where rows. This allows us to write the test statistic as:

$$
\begin{aligned}
\alpha^\top \Bigg( & L(Y^{\mathrm{Te}}, Y^{\mathrm{Te}}) + 2\mathrm{diag}(\mathbf{e})K(X_0^{\mathrm{Tr}}, X^{\mathrm{Te}})^\top \mathbf{W}_0 L(Y_0^{\mathrm{Tr}}, Y^{\mathrm{Te}}) - 2\mathrm{diag}(\mathbf{1} - \mathbf{e})K(X_1^{\mathrm{Tr}}, X^{\mathrm{Te}})^\top \mathbf{W}_1 L(Y_1^{\mathrm{Tr}}, Y^{\mathrm{Te}}) \\
& + \mathrm{diag}(\mathbf{e})K(X_0^{\mathrm{Tr}}, X^{\mathrm{Te}})^\top \mathbf{W}_0 L(Y_0^{\mathrm{Tr}}, Y_0^{\mathrm{Tr}})\mathbf{W}_0 K(X_0^{\mathrm{Tr}}, X^{\mathrm{Te}})\mathrm{diag}(\mathbf{e}) \\
& + \mathrm{diag}(\mathbf{1} - \mathbf{e})K(X_1^{\mathrm{Tr}}, X^{\mathrm{Te}})^\top \mathbf{W}_1 L(Y_1^{\mathrm{Tr}}, Y_1^{\mathrm{Tr}})K(X_1^{\mathrm{Tr}}, X^{\mathrm{Te}})\mathbf{W}_1\mathrm{diag}(\mathbf{1} - \mathbf{e}) \\
& - 2\mathrm{diag}(\mathbf{e})K(X_0^{\mathrm{Tr}}, X^{\mathrm{Te}})^\top \mathbf{W}_0 L(Y_0^{\mathrm{Tr}}, Y_1^{\mathrm{Tr}})K(X_1^{\mathrm{Tr}}, X^{\mathrm{Te}})\mathbf{W}_1\mathrm{diag}(\mathbf{1} - \mathbf{e}) \Bigg) \alpha.
\end{aligned}
$$

## B.2  Derivation of $\widehat{\mathrm{MMD}}_{\mathrm{DR}}\Big[\left\{Y(1)\mid T=0\right\},\left\{Y(0)\mid T=0\right\},\mathcal{H}_{\mathcal{Y}}\Big]$

We now derive the closed form of the estimator based on the effect of treatment on the treated for the case of a binary treatment. Again we begin by deriving a simpler form of the difference between mean embeddings:

$$
\begin{aligned}
\hat{\mu}_{\{Y(1)|T=0\}} - \hat{\mu}_{\{Y(0)|T=0\}} &= \hat{\mu}_{\{Y(1)|T=0\}} - \hat{\mu}_{\{Y|T=0\}} \\
&= \frac{1}{n_0}\sum_{i=1}^{n}\left(t_i\hat{w}(x_i,1)\left(\ell(y_i,\cdot)-\hat{r}(x_i,1)\right)+(1-t_i)\,\hat{r}(x_i,1)\right)-\frac{1}{n_0}\sum_{i=1}^{n}(1-t_i)\,\ell(y_i,\cdot) \\
&= \frac{1}{n_0}\sum_{i=1}^{n}\left(t_i\hat{w}(x_i,1)-(1-t_i)\right)\left(\ell(y_i,\cdot)-\hat{r}(x_i,1)\right) \\
&= \frac{1}{n_0}\sum_{i=1}^{n}\left(\frac{t_i-e(x_i)}{e(x_i)}\right)\left(\ell(y_i,\cdot)-\hat{r}(x_i,1)\right)
\end{aligned}
$$

Now we have that $\hat{r}(x_i,1)=\mu_{Y|X=x,T=1}=\mathbf{k}_1^{\top}(x)\mathbf{W}_1\mathbf{l}_1$ and we also let:

$$
\beta(x_i)=\frac{t-e(x_i)}{e(x_i)}
$$

Then we can write the full test stat as:

$$
\begin{aligned}
\left\|\hat{\mu}_{\{Y(1)|T=0\}}-\hat{\mu}_{\{Y(0)|T=0\}}\right\|^2 &= \frac{1}{n_0^2}\sum_{i,j}\beta(x_i)\beta(x_j)\left\langle \ell(y_i,\cdot)-\mathbf{k}_1^{\top}(x_i)\mathbf{W}_1\mathbf{l}_1,\ell(y_j,\cdot)-\mathbf{k}_1^{\top}(x_j)\mathbf{W}_1\mathbf{l}_1\right\rangle \\
&= \frac{1}{n_0^2}\sum_{i,j}\beta(x_i)\beta(x_j)\big(\ell(y_i,y_j)-\mathbf{k}_1^{\top}(x_i)\mathbf{W}_1\mathbf{l}_1(x_j)-\mathbf{k}_1^{\top}(x_i)\mathbf{W}_1\mathbf{l}_1(x_i)+ \\
&\qquad\qquad\qquad\qquad + \mathbf{k}_1^{\top}(x_j)\mathbf{W}_1\mathbf{L}_{1,1}\mathbf{W}_1\mathbf{k}_1^{\top}(x_j)\big),
\end{aligned}
$$

giving the full test statistic as:

$$
\beta^{\top}\left(L(Y^{\mathrm{Te}},Y^{\mathrm{Te}})-2K(X_1^{\mathrm{Tr}},X^{\mathrm{Te}})^{\top}\mathbf{W}_1 L(Y_1^{\mathrm{Tr}},Y^{\mathrm{Te}})+K(X_1^{\mathrm{Tr}},X^{\mathrm{Te}})^{\top}\mathbf{W}_1 L(Y_1^{\mathrm{Tr}},Y_1^{\mathrm{Tr}})\mathbf{W}_1 K(X_1^{\mathrm{Tr}},X^{\mathrm{Te}})\right)\beta.
$$

# C  Simulation Details

For the simulation in Section 5.1 we select:

$$
\begin{aligned}
\mathbf{a} &= [0.1,0.2,0.3,0.4,0.5,0.1,0.2,0.3,0.4] \\
\mathbf{b} &= [0.5,0.4,0.3,0.2,0.1,0.4,0.3,0.2,0.1] \\
\sigma &= 0.2 \\
\beta &= 3,
\end{aligned}
$$

and for the second experiment we let:

$$
\begin{aligned}
f_0(x) &= x_1 & f_1(x) &= x_1^2 \\
\alpha &= 0.3 & \sigma &= 0.2.
\end{aligned}
$$

## C.1  Fit Experiment Plots

Here we include the propensity plots for the simulated data in the fit plots:

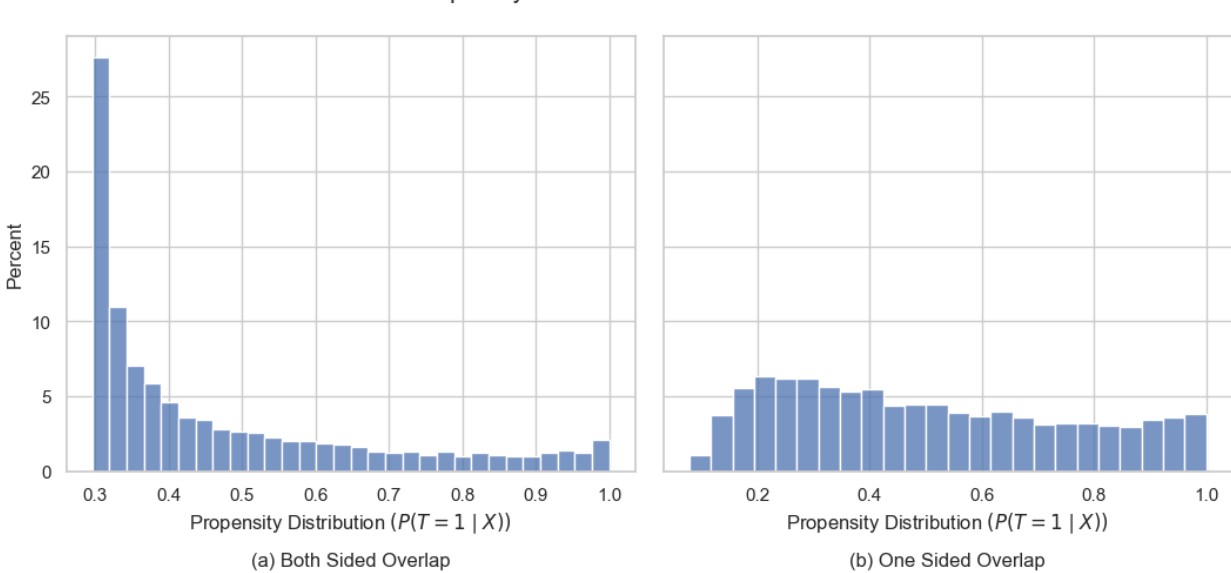

We can see in the second plot with one sided overlap a much higher proportion of datapoints are concentrated at extreme propensities.

## C.2 Simulated Distributional Test Plots

Here we include some plots for the distribution in the experiments testing for causal effects on simulated data:

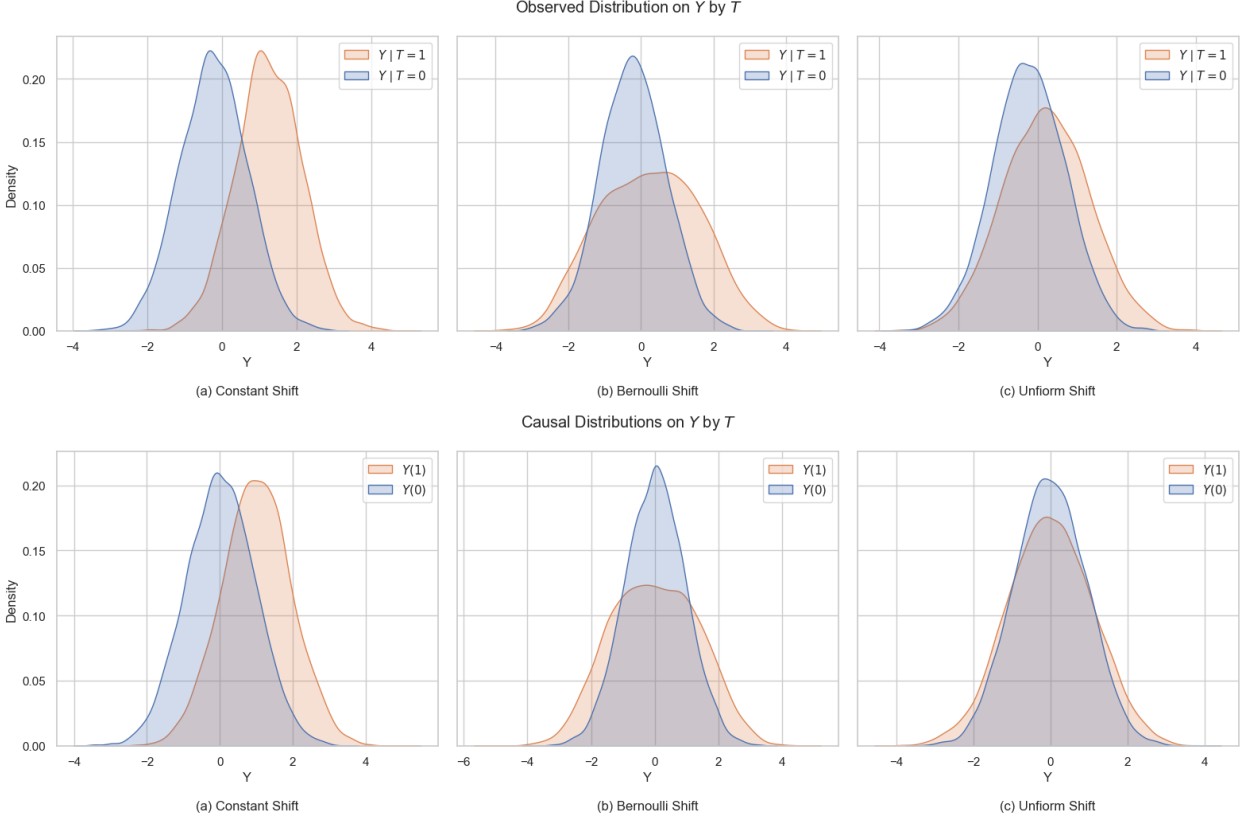

And also the distribution over propensity scores:

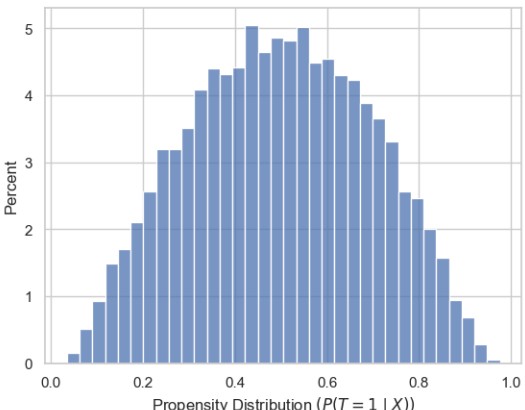

