# OpenReview forum: "Doubly Robust Kernel Statistics for Testing Distributional Treatment Effects"
_TMLR — Accepted by TMLR_

### Review · Reviewer_YytB · 2024-02-27

**Summary Of Contributions:**

The paper studies the problem of testing for distributional causal effects. There are two main contributions of the paper:

1.	The authors make use of the counterfactual mean embeddings framework and propose an estimator for the distributional embeddings. Unlike previous literature, the proposed estimator is doubly robust. The authors establish theoretical properties for the proposed estimator. This estimator can serve as a test statistic in the testing stage.

2.	The authors propose a permutation test that utilizes the above estimator as the test statistic. In particular, the authors employ a smart approach that helps alleviate the computational challenges in permutation tests.

**Audience:**

Yes

**Claims And Evidence:**

Yes

**Requested Changes:**

I'm going to list my questions and comments here. I'm generally positive about the paper, and all the comments and questions below are intended to strengthen the work from my perspective.

1.	More background in Section 1: It would be helpful if the authors could provide more background before delving into the technical details in the first section. Inform readers who are not familiar with causal inference more about why this problem is significant, and why previous methods may not be adequate.
2.	Page 4, the third equation from the bottom, contains a typo: it should be Y(0)|T=1 instead of Y(1)|T=0.
3.	Around equation (3) on page 7: I particularly like this idea. It is very nice and effectively addresses the issue. It would be helpful if the authors could provide more intuition about what the procedure is doing. For example, explaining why choosing N = 5 in the experiment does not undermine the power of the test would be beneficial for the readers.
4.	The goal of developing a doubly robust procedure: It seems that the proposed doubly robust estimator would be particularly advantageous when the propensity score is mis-specified. However, in the context of testing, the matching step would also fail, or at least not perform well, if you use propensity score matching; then the test becomes invalid. Conversely, if the propensity score is correctly specified, then the doubly robust estimator would perform similarly to the original estimator. What is the advantage of developing a doubly robust estimator in this scenario? Furthermore, achieving double robustness reduces the effective sample size due to sample splitting. It would be beneficial if the authors could clarify these points and/or provide additional simulation studies to justify their findings.
5.	Figure 4 (b) and (c): The size of the test seems to exceed the 0.05 threshold, especially for the red line. Are the authors concerned about the potential for an inflated type-I error rate?
6.	Extreme data points: On Page 10, the authors state, "Therefore, to prevent problems with extreme propensity scores, we remove any data for which e(x, t) < 0.03 for any value of t." Is this approach recommended for practitioners in general?
7.	Image or graph simulation: In the first paragraph, the authors motivate the problem by stating, "This can be especially useful when the target variable is high-dimensional or structured (e.g., an image or a graph)." It is somewhat disappointing that the simulation studies later in the paper are confined to simpler scenarios. It would enhance the paper if the authors could include simulations on image or graph data, or provide stronger motivation for these simpler settings.

**Strengths And Weaknesses:**

Strengths And Weaknesses Strengths:

1.	The paper addresses an interesting and important question.

2.	The paper is written in a clear and engaging manner.

3.	The proposed method appears to perform well both theoretically and empirically.
Weaknesses: See next section for more details.

---

> ### Author Response · Authors · 2024-03-07
> **Response to Reviewer YytB**
>
> We thank the reviewer for their positive feedback on our paper, and for their constructive feedback. Responding to the requested changes in order:
>
>
> 1. To incorporate reviewer feedback we will better motivate the problem in the introduction.
> 2. We thank the reviewer for noticing, we will make the change.
> 3. We thank the reviewer for recognising the positives with of the permutation procedure. We will add more intuition to this section to explain the functionality, as well as adding further experiments to showcase varying $N$. From the work we have done so far, we found that increasing $N$ did not improve the power of our tests, but did improve calibration. $N=5$ was chosen to trade off this calibration and computational cost.
> 4. We would like to begin by pointing out that there are applications of counterfactual mean embeddings outside of testing, for which our estimators could be used; for example sampling from the counterfactual distribution. Within the context of testing, whilst we have used propensity score matching their have been numerous other approaches considered in the literature which do not rely on the propensity score model. These could be used within our test, and would produce a valid test even if the propensity score is misspecified, so long as the matching method is correctly specified. When running these experiments we found little difference in performance and so did not include them, however we would be happy to put them in an appendix if the reviewer requests. Finally, we would like to add that our statistics could be used in recent permutation free approaches [1], where the distributional effect of treatment on the treated has yet to be considered.
> 5. We have run large scale experiments verifying the calibration of our approach and so are not concerned about inflated type 1 error. We can include these results in an Appendix to further demonstrate the correctness of our procedure.
> 6. This is quite standard practice in the literature on doubly robust estimation as otherwise we end up with very high variance estimators, we will add a discussion of this and relevant references to the manuscript.
> 7. While the test is applicable to general domains, we kept the experimental evaluation within the controlled contexts where it is easier to study effects of individual methodological choices. Further, their are currently few high quality open source datasets which are suitable for causal inference and have structured outcomes. For both these reasons we remained in relatively simple settings. However for future work, with available datasets, our tests could be applied to functional [2] or graphical data.
>
> [1] - An Efficient Doubly-Robust Test for the Kernel Treatment Effect. Diego Martinez-Taboada, Aaditya Ramdas, Edward H. Kennedy: https://arxiv.org/abs/2304.13237
> [2] - A Kernel Two-Sample Test for Functional Data. George Wynne, Andrew B. Duncan : https://arxiv.org/abs/2008.11095.

---

### Review · Reviewer_8X9u · 2024-03-01

**Summary Of Contributions:**

This paper proposes a novel framework, based on Counterfactual Mean Embeddings, for testing distributional causal effects in a diverse array of circumstances, in which a doubly robust estimator is designed from semi-parametric statistics. It is interesting to see that the estimator could represent causal distributions within Reproducing Kernel Hilbert Spaces (RKHS). In theoretical section, they analyze the estimator from kernel space, and prove that the proposed estimator can retain the doubly robust property and have improved convergence rates. The extensive experimental results verify the effectiveness of the proposed framework in estimating distributional casual effects.

**Audience:**

Yes

**Broader Impact Concerns:**

nan

**Claims And Evidence:**

Yes

**Requested Changes:**

1. The authors introduce a one-sided overlap assumption, but in theorem 2, they did not amylase the connection with MMD. Is it helpful to obtain convergence rate or lower the convergence rate? . Because, it requires e(x,t) > \epsilon, it seems like the strict overlap satisfied that?
2. The authors claim that the proposed estimator is doubly robustness due to the properties of semi-parametric statistics, But it is little difficult to see the connections between them?
3, The experiments are not solid from my perspective, especially, it lacks robustness certification experiments.
Typos:
 analyse -> analyze
Kernelized-> kernelized
Parametric-> parametric
Permuations-> permutations
…
Please proofread the paper again and correct the typos.

**Strengths And Weaknesses:**

1. In order to estimate the distributional casual effects, the authors introduce a novel estimator to improve the robustness of casual effects estimation. Different to previous works, the estimator is based on semi-parametric statistics.
2. The theoretical analysis proved the double robustness properties of the proposed framework. Based on that, the proposed estimator can converge to a correct value.
3. In order to verify the robustness of the proposed estimator, the authors apply it to permutation testing. The results are consistent to the expectation, and agree with the analysis in theoretical section.
4. they conduct extensive experiments on synthetic, semi-synthetic and real-world datasets. The experimental results indeed prove their idea. The detailed implementation are presented at Github, which is helpful to causal community.

---

> ### Author Response · Authors · 2024-03-07
> **Response to Reviewer 8X9u**
>
> We would like to thank the reviewer for their feedback and for recognising the extensiveness of our experiments in proving the validity of our method. We would like to make the following points in relation to the reviewers suggested changes, in the order they appear:
>
> 1. We will improve our discussion of the overlap conditions in the text below theorem one and theorem two. The main point is that both-sided strict overlap is required for the estimator in theorem one to converge, whereas only one sided strict overlap is needed in theorem 2. Therefore the distributional effect of treatment on the treated will converge in scenarios where the distributional average treatment effect will not. This can be seen in our experiments in Figure 3.
>
> 2. To clarify the relationship between all the terms, the original doubly robust estimator of the causal mean is one of the key methods in the semi-parametric literature. It has the doubly robust property that it will converge if either of the underlying models converge. In our case, the estimator we propose is heavily inspired by this original doubly robust estimator, which is the link to semi-parametric statistics we referred to. We will make this more clear in the introduction.
>
> 3. In terms of studying the robustness, our experiments in Figure 2. demonstrate the robustness of the statistics to misspecification of the propensity score, Figure 3. shows the robustness of a portion of the statistics to overlap violations. Finally our experiments on simulated and semi-synthetic datasets demonstrates the functionality of our tests in a wide variety of settings.
>
> Finally thank you for pointing out all typos, we will make the changes.

---

### Review · Reviewer_hYPV · 2024-03-01

**Summary Of Contributions:**

Post rebuttal update: The authors' explanations and/or proposed changes largely addressed my concerns. I changed the answer of "Claims And Evidence" to "Yes".

The paper combines Counterfactual Mean Embeddings (CMEs) in machine learning and the classical ideas of doubly robust estimators in causal inference. It uses kernel mean embeddings to build kernelized versions of the classical doubly robust estimators. As to testing the null hypothesis of no treatment effects, the paper uses the recent result of Ramdas et al. (2023) to reduce the computational load.

**Audience:**

Yes

**Claims And Evidence:**

Yes

**Requested Changes:**

## Critical

The test is of limited novelty and seems to be an afterthought. Despite the statement “this procedure could be applied to efficiently use other trainable test statistics within permutation testing”, my reading (of the main text and A.3) is that, because the proposed statistics are computationally heavy to apply to naive permutation tests, the authors use Ramdas et al. (2023)’s idea which itself can be applied to general test statistics in permutation testing. If this is a misunderstanding, the authors need to discuss the original work and compare the proposed test to it. Otherwise, I request revising the claims such as “This *leads* to *new* permutation-based tests….”, “We propose a new permutation approach…” and “this is the first time this permutation strategy has appeared in the literature” to more humble ones. I also suggest changing the title to “Doubly Robust Kernel Statistics for Distributional Treatment Effects”, without stressing the test.

## Good to have (or some important questions)

The definition of (one-sided) overlap is a bit deviated from the standard and would confuse inexperienced readers. For overlap, the def contains unspecified $t$, and, if we understand this as “for both $t=0,1$”, then there is no need to require “$<1$”. Also, there are 2*2=4 kinds of “one-sided overlap”, but what is used in the paper are “$e(x,0)>0$” and “$e(x,1)>0$”; I would call them “treatment/control group positivity” respectively (the term “overlap” loses its intuitive meaning if we consider only one of the group).

In Theorem 1 and 2, why do we need “$\gamma_{e,n}=o(1)$ and $\gamma_{r,n}=o(1)$”? Doesn’t this mean “both estimators converge“? The proofs seem to just say “the rate is controlled by the product rate of the two estimators” and we can have the conclusion without saying anything about the rates of the two estimators?

I am not sure why the proposed estimators “have theoretically improved convergence rates when compared to the previous counterfactual mean embedding estimators”? As indicated in the paper, previous CME estimators also converge at the parametric rate, and, according to Theorem 1 and 2, the proposed estimators achieve no better rates than this (although they have double robustness)?

Approximate validity under inexact matching. As far as I understand, in Simulation 3, the matching is exact because both the model and truth are logistic. So I guess the semi-synthetic data do not use the logistic function to generate the treatment and thus the matching is inexact? Even if so, it needs to be explained clearly and the data-generating process should be given in the Appendix.

## Minor

The symbol $S$ is overloaded for both a permutation group and a statistic. I suggest the symbol $T$ for the latter.

In the caption of Fig 4, “introduces a shift in the” → “introduces a shift in the distribution”.

**Strengths And Weaknesses:**

## Strengths

The extension of CMEs to double robustness is a natural and valuable contribution.

Theoretical and experimental analysis is relatively solid.

## Weaknesses

The theoretical analysis seems to be an adaptation of previous works.

The permutation test does not follow naturally from the kernel statistics but is directly adapted from Ramdas et al. (2023) as a way to reduce the computation. It is more like a practical compromise than a novelty. See Requested Changes for details.

---

> ### Author Response · Authors · 2024-03-07
> **Response to Reviewer hYPV**
>
> We thank the reviewer for their time and feedback on our work. We would like to begin by responding to the reviewers most pressing points:
>
> # Testing and Permutation Procedure
>
> A key concern of the reviewer is the novelty of our testing procedure in relation to [1], so we would like to clarify our contribution: [1] provides a general way to produce a valid permutation test when using an arbitrary subset, $S$, of the permutation group. They propose an alternative way to calculate the p-value which compensates for using this arbitrary subset, and prove that the compensation leads to a valid p-value. In our work, we construct a specific subset $S$, so that we can avoid having to retrain our models for every permutation. This is important as without this the test would be computationally infeasible. We then apply the result of [1] to construct the p-value of the test and prove it is valid. So, to conclude, our contribution here is a methodological one, specifically the construction of the set $S$. After this we apply [1] to prove its theoretical validity.
>
> We will add further clarification on this to the paper, with references to [1] so that our precise contribution is clear. Further, we will dial down claims as follows:
>
> - Changing "This leads to new permutation-based tests for distributional causal effects"  to "This leads us to propose new permutation-based tests for distributional causal effects".
>
> - Removing the paragraph beginning with " “this is the first time this permutation strategy has appeared in the literature".
>
> - Changing "We propose a new permutation approach which allows for doubly robust trainable statistics to be used within permutation testing and prove that these tests are valid." to "We apply recent results on generalised permutation tests [1] to create a bespoke permutation procedure which reduces the computational overhead of retraining the statistic for each permutation."
>
> Finally, whilst we appreciate the reviewers suggestion for a title change, we would prefer to stick to our original title. This is because testing is one of the main applications of kernel mean embeddings. Further, testing for distributional treatment effects is a main application of the counterfactual mean embedding paper [2], as well as in subsequent and strongly related work [2,3,4].
>
> # Other Changes:
> Finally, commenting on the reviewers additional points in the order they appeared:
>
> 1. We thank the reviewer for pointing out the ambiguity regarding $t$. We will change this so that overlap is defined as:
> $$
>     e(X,t) > 0,
> $$
> where we have treatment positivity if this holds for $t=1$, control positivity if this holds for $t=0$, and one sided overlap if we have only one of treatment of control positivity.
> 2. We thank the reviewer for point this out, we will change this to $O(1)$ for both $\gamma_{e,n},\gamma_{r,n}$.
>
> 3. The rate proved in [2] only holds when the propensity score is correctly specified, otherwise our proof demonstrates it will converge at rate: $O_P( max [ n^{-\frac{1}{2}},\gamma_{e,n}]  )$ .
> On the other hand, our estimators will converge at rate $O_P( max [ n^{-\frac{1}{2}},\gamma_{e,n} \gamma_{r,n}]  )$. As the conditional mean embedding will always converge by [5], we have that $\gamma_{r,n}$ will tend to zero. Therefore, as long as $\gamma_{e,n}$ is not $O(n^{-\frac{1}{2}})$, our estimator will have a strictly better rate than those in [2]. We will add this discussion to the paper to increase clarity.
> 4. Exact matching does not relate to permuting with the exact propensity score, but only permuting for individuals who have exactly the same true propensity. When the covariates are continuous, it is highly unlikely that two individuals will have the same true propensity, therefore even when the propensity model is correct we will be performing an inexact matching. However, we will more clearly explain this point in the manuscript and add the data generating process in the Appendix.
> We will also make the small changes pointed out by the reviewer.
>
> [1] - Permutation tests using arbitrary permutation distributions. Aaditya Ramdas, Rina Foygel Barber, Emmanuel J. Candes, Ryan J. Tibshirani : https://arxiv.org/abs/2204.13581
> [2] - Counterfactual Mean Embeddings. Krikamol Muandet, Motonobu Kanagawa, Sorawit Saengkyongam, Sanparith Marukatat: https://arxiv.org/abs/1805.08845
> [3] -A kernel two-sample test with selection bias. Alexis Bellot, Mihaela van der Schaar: https://proceedings.mlr.press/v161/bellot21b.html
> [4] - An Efficient Doubly-Robust Test for the Kernel Treatment Effect. Diego Martinez-Taboada, Aaditya Ramdas, Edward H. Kennedy: https://arxiv.org/abs/2304.13237
> [5]- Optimal Rates for Regularized Conditional Mean Embedding Learning. Zhu Li, Dimitri Meunier, Mattes Mollenhauer, Arthur Gretton: https://arxiv.org/abs/2208.01711

---

> > ### Comment · Reviewer_hYPV · 2024-03-20
> > **Thanks for reply**
> >
> > The authors' explanations and/or proposed changes largely addressed my concerns. I will add only one point: "This leads us to propose new permutation-based tests" in the Abstract reads still as if the "doubly robust property" and/or "improved convergence rates" lead to the tests, but this is not the case.

---

### Decision · Action_Editor_qJiq · 2024-05-26

**Recommendation:** Accept with minor revision

**Comment:**

After discussion, this paper is acceptable with minor revisions, outlined below.
1. Sec 2 presumes the reader is already familiar with potential outcomes and causal terminology. Please either add a short explainer with some more background on causal inference for those not familiar with the field, or add an appropriate reference for the reader to get up to speed.
2. Please correct and clarify the following: "This leads us to propose new permutation-based tests" in the Abstract reads still as if the "doubly robust property" and/or "improved convergence rates" lead to the tests, but this is not the case.
3. Please carefully check and fix all typos (all reviewers had minor suggestions in their reviews)

Congratulations on a well-written paper!

**Audience:**

Understanding causal inference in general settings, such as detecting higher-order distributional effects in structured data, is an important problem that is clearly relevant to the TMLR audience.

**Claims And Evidence:**

All reviewers agreed that the paper is well-written and supported by sufficient evidence. During the discussion, the authors successfully clarified the concerns raised by one reviewer, and there is a consensus to accept the paper with minor revisions.